# R-KV: Redundancy-aware KV Cache Compression for Reasoning Models

**Zefan Cai**[1✉]**, Wen Xiao**[2✉]**, Hanshi Sun**[3]**, Cheng Luo**[4]**, Yikai Zhang**[1]**, Ke Wan**[5]**, Yucheng Li**[6]**,
Yeyang Zhou**[5]**, Li-Wen Chang, Jiuxiang Gu**[7]**, Zhen Dong**[8]**, Anima Anandkumar**[4]**,
Abedelkadir Asi**[2]**, Junjie Hu**[1✉]
[1]University of Wisconsin - Madison [2]Microsoft [3]Carnegie Mellon University
[4]California Institute of Technology [5]University of California - San Diego [6]University of Surrey
[7]Adobe [8]University of California - Berkeley
`https://zefan-cai.github.io/R-KV.page/`
`https://github.com/Zefan-Cai/R-KV`

## Abstract

Reasoning models have demonstrated impressive performance in self-reflection and chain-of-thought reasoning. However, they often produce excessively long outputs, leading to prohibitively large key-value (KV) caches during inference. While chain-of-thought inference significantly improves performance on complex reasoning tasks, it can also lead to reasoning failures when deployed with existing KV cache compression approaches. To address this, we propose **R**edundancy-aware **KV** Cache Compression for **R**easoning models (**R-KV**), a novel method specifically targeting redundant tokens in reasoning models. Our method preserves nearly 100% of the full KV cache performance using only 10% of the KV cache, substantially outperforming existing KV cache baselines, which reaches only 60% of the performance. Remarkably, R-KV even achieves 105% of full KV cache performance with 16% of the KV cache. This KV-cache reduction also leads to a 90% memory saving and a 6.6× throughput over standard chain-of-thought reasoning inference. Experimental results show that R-KV consistently outperforms existing KV cache compression baselines across two mathematical reasoning datasets.

## 1 Introduction

Recent advancements in large language models (LLMs) have demonstrated remarkable capabilities in complex reasoning and self-reflection. However, reasoning models (e.g., DeepSeek-R1 [1]) exhibit a critical deployment challenge: their tendency to produce excessively lengthy and redundant reasoning traces results in unsustainable memory demands [2], primarily due to the rapid growth of the key-value (KV) cache during autoregressive generation. For instance, a DeepSeek-R1-Distill-Llama-8B model may generate 32K tokens to solve a complex math problem, consuming 15.5GB of memory to load the model weight and 4.1GB of memory to store the KV cache. This paradigm of long chain-of-thought (CoT) reasoning generation necessitates the development of KV cache compression.

Outputs from current reasoning models, especially during complex chain-of-thought generation, are fundamentally marked by pervasive redundancy. This inherent characteristic means they are often filled with superfluous content, including unnecessary reflections, iterative re-evaluations, and verbose self-dialogue, all of which add little new semantic value while significantly inflating the

---

✉ Corresponding to Zefan Cai `zefncai@gmail.com`, Wen Xiao `wxiao@microsoft.com` and Junjie Hu `junjie.hu@wisc.edu`

39th Conference on Neural Information Processing Systems (NeurIPS 2025).

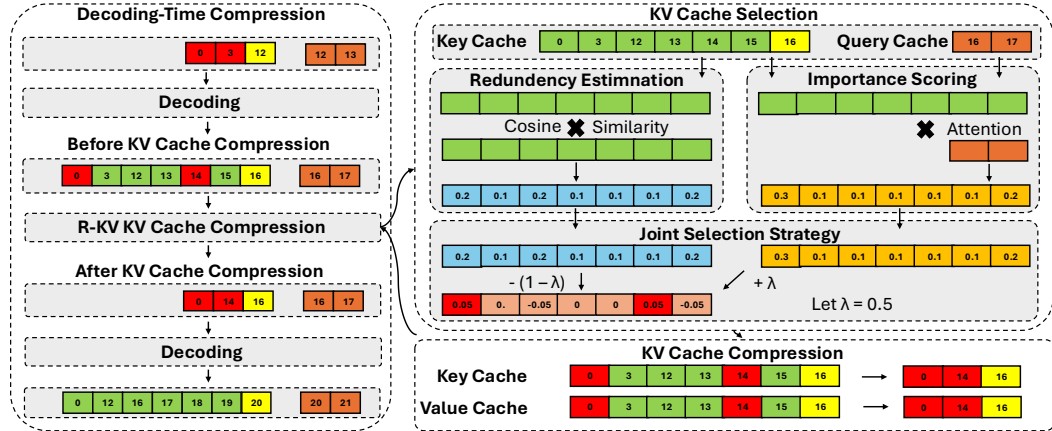

Figure 1: R-KV: (1) Decoding-Time Compression (§3.1); (2) KV Cache Selection with Importance and Redundancy Estimation (§3.2, §3.3) ; (3) KV Cache Compression by joint selection (§3.4).

length of the generation beyond what is needed for concise, effective reasoning. Our analysis (§2.1) shows that over half of the tokens in R1's reasoning chains contribute minimally to task performance, indicating that repetitive self-verification steps or intermediate calculations could be substantially condensed by KV cache compression methods without compromising reasoning accuracy.

However, existing KV cache compression works [3, 4, 5, 6, 7] primarily handle long input prompts but do not explore extensively for long generation outputs. Furthermore, based on our observation (§2.2), standard KV-cache compression methods that rely on simple attention-based importance filtering often fail because the repetitive sections generate high attention signals for themselves. Naively pruning tokens with "low attention weight" may remove crucial but scattered bits of reasoning, or over-retain duplicative self-reflections that appear to have high attention. This observation motivates our exploration of redundancy-aware compression strategies, which selectively retain "important and non-repetitive context" during decoding to preserve the model's critical reasoning ability.

In this work, we propose **R**edundancy-aware **KV** cache compression for reasoning models (i.e., **R-KV**). Our approach consists of three key components: (1) an attention-based importance scoring mechanism that selects critical tokens for retention, (2) a dynamic redundancy scoring mechanism that identifies repetitive tokens through real-time analysis of key vectors, and (3) a joint eviction mechanism that balances both redundancy and importance to optimize cache efficiency.

In our experiments on popular math reasoning benchmarks (§4), by selectively retaining **only 10-34%** of the original KV cache, R-KV achieves comparable performance parity with the uncompressed reasoning model, outperforming state-of-the-art compression baselines with only **60%** of the performance. Remarkably, R-KV even achieves **105%** accuracy of the full KV baseline with around **16%** of the KV cache using DeepSeek-R1-Distill-Llama-8B on the AIME-24 dataset.

This advancement addresses a fundamental tension in deploying state-of-the-art LLMs—balancing reasoning capabilities with practical memory constraints. Our contributions extend beyond technical optimization: we provide systematic evidence that redundancy in CoT generation can be strategically compressed without compromising reasoning abilities. As a training-free and model-agnostic method, R-KV can be used in the rollout process in reinforcement learning (RL) and LLM serving.

## 2 Observation

### 2.1 Redundancy in Reasoning Models

As noted in [2], reasoning models often generate a detailed chain of thoughts and multiple reflection steps, resulting in significantly longer responses than standard models. Figure 2 shows that both reasoning models (i.e., DeepSeek-R1-Distill-Llama-8B, DeepSeek-R1-Distill-Qwen-7B and DeepSeek-R1-Distill-Qwen-14B) generate more than $8\times$ longer generation output compared to the ground truth on two popular math reasoning datasets. However, not all of the additional tokens contribute meaningful content, as much of the decoded context is dominated by repetition. Figure 2

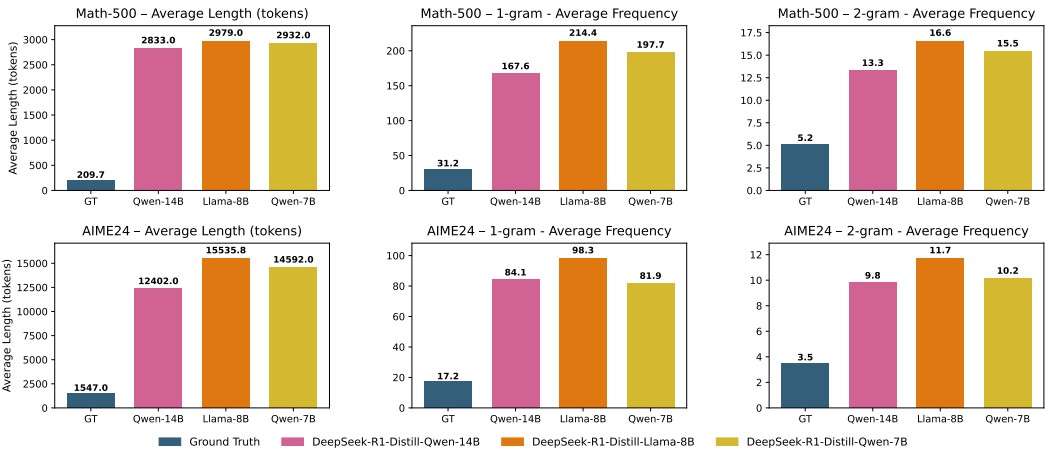

Figure 2: Comparison of generation length and average 1-/2-gram frequency for reasoning models and ground truth of MATH-500 [8] and AIME 2024 [9]. . Reasoning models generate substantially longer responses with 8-14× more tokens, and show higher word repetition with 5-7× higher frequency.

also shows that the average frequency of 1- to 2-grams is consistently higher in the generation output of reasoning models than in ground truth, indicating greater repetitions in the generated outputs of reasoning models.

## 2.2 Failure of Existing KV Compression Methods to Handle Redundancy

Most existing KV cache compression methods prioritize token selection based primarily on tokens' contextual importance, typically measured through attention scores between key and query tokens [3, 4]. While this approach effectively retains critical context, it fails to account for redundancy—particularly problematic in reasoning models. In such models, we find that repetitive content often receives disproportionately high attention scores, as it closely mirrors previously generated repetitive text. As a result, redundant tokens are excessively retained, unnecessarily inflating the KV cache size without providing additional meaningful new information. In Figure 3, we visualize the cached tokens (inside red boxes) selected by a popular attention-based KV cache method (i.e., SnapKV), showing many repetitions related to self-reflection and conclusion to the final answer.

Figure 3: KV selected by SnapKV. SnapKV suffers from redundancy in reasoning models. Black tokens are not selected by SnapKV; brighter colors reflect higher attention scores. Blue tokens are omitted output.

# 3 Redundancy-aware KV Cache Compression (R-KV)

To address the redundant thinking issue, we propose a *redundancy-aware decoding-time KV cache compression method* (**R-KV**) that explicitly targets the compression of redundant tokens in reasoning models. Our approach balances *importance* and *non-redundancy* in token selection, ensuring that KV cache storage is allocated to both highly informative and diverse content. By incorporating

redundancy estimation into the selection process, our method effectively mitigates unnecessary KV cache growth while preserving the model's reasoning capabilities.

Specifically, R-KV consists of three key components: (1) an *importance scoring mechanism* (§3.2) leveraging attention weights, (2) a *redundancy estimation mechanism* (§3.3) based on semantic similarity of key vectors, and (3) a *joint selection strategy* (§3.4) that optimizes cache efficiency by balancing redundancy and importance.

## 3.1 Decoding-time Compression

Different from existing KV cache compression methods[3, 5, 4] that focus on the *prefilling stage* to manage long-context inputs, our R-KV focuses on the *decoding stage* for reasoning models—a distinctive setting where the generated output is significantly longer than the prompt.

Specifically, R-KV allocates memory for two components: a cache of budget size $B_{\text{budget}}$ to store retained KV tokens, and a buffer of size $B_{\text{buffer}}$ for newly generated text tokens. The total memory requirement is thus $B_{\text{total}} = B_{\text{budget}} + B_{\text{buffer}}$. After the model generates each fixed-length text segment in the buffer, R-KV performs KV cache compression. At the end of each text segment, the last $\alpha$ tokens are always retained in the cache as **observation tokens**, following prior work [3]. Next, we concatenate the existing $B_{\text{budget}}$ tokens in the cache with the first $B_{\text{buffer}} - \alpha$ tokens in the buffer, resulting in $n = B_{\text{budget}} + B_{\text{buffer}} - \alpha$ candidate KV tokens. Each candidate is assigned a selection score (§3.4), and we select the top $k = B_{\text{budget}} - \alpha$ tokens to fit in the rest of the cache budget, in addition to the $\alpha$ observation tokens. This process compresses the KV cache while preserving critical context, enabling efficient memory utilization during autoregressive decoding.

## 3.2 Importance Scoring via Attention Weights

Following attention-based methods (e.g., SnapKV [3], PyramidKV [5]), R-KV estimates token importance using attention weights, leveraging the intuition that tokens receiving higher attention contribute more to decoding and are thus more critical for preserving model performance. Specifically, we compute each key token's attention scores received from the last $\alpha$ **observation tokens** during decoding. In addition to the standard multi-head attention mainly adopted by the prior works [3], we also propose the importance score estimation using the grouped-query attention. Below, we detail the estimation on top of these two popular attention mechanisms used by current LLMs.

**Multi-Head Attention (MHA).**  Given the last $\alpha$ observation tokens as query $\boldsymbol{Q}^h \in \mathbb{R}^{\alpha \times d}$ and $n$ key states $\boldsymbol{K}^h \in \mathbb{R}^{n \times d}$ for each attention head $h$, the attention scores $\boldsymbol{A}^h \in \mathbb{R}^{\alpha \times n}$ are computed as:

$$\boldsymbol{A}^h = \text{softmax}(\boldsymbol{Q}^h \cdot (\boldsymbol{K}^h)^\top / \sqrt{d}). \tag{1}$$

**Grouped-Query Attention (GQA).**  In GQA, each key/value head $h$ is shared among a group of $G$ distinct query heads indexed by $g \in [0, G)$. Correspondingly, we denote the shared key/value states as $\boldsymbol{K}^h, \boldsymbol{V}^h \in \mathbb{R}^{n \times d}$, and the $G$ query states as $\boldsymbol{Q}^{h,0}, \ldots, \boldsymbol{Q}^{h,G-1} \in \mathbb{R}^{\alpha \times d}$ within the head group indexed by $h$, where $n$ is the number of key/value states, $d$ is the head hidden dimension. The attention score for each of the $G$ query heads within the group is computed as:

$$\boldsymbol{A}_{\text{group}}^{h,g} = \boldsymbol{Q}^{h,g} \cdot (\boldsymbol{K}^h)^\top / \sqrt{d} \in \mathbb{R}^{\alpha \times n}, \quad \text{for } g = 0, \ldots, G-1. \tag{2}$$

These $G$ individual matrices are then aggregated into a single consolidated matrix $\boldsymbol{A}_{\text{group}}^h$ for the head group $h$ using a max-pooling operation across the group dimension. The final attention weight $\boldsymbol{A}^h$ for the head group $h$ is then obtained by renormalizing $\boldsymbol{A}_{\text{group}}^h$ along the key token dimension.

$$\boldsymbol{A}_{\text{group}}^h = \text{maxpool}\left(\left[\boldsymbol{A}_{\text{group}}^{h,0}, \ldots, \boldsymbol{A}_{\text{group}}^{h,G-1}\right]\right) \in \mathbb{R}^{\alpha \times n}, \quad \boldsymbol{A}^h = \text{softmax}\left(\boldsymbol{A}_{\text{group}}^h\right) \in \mathbb{R}^{\alpha \times n} \tag{3}$$

**Stabilization and Importance Estimation.**  We use $\boldsymbol{A}^h$ hereafter to denote the attention weights calculated using either MHA or GQA. Note that the per-token importance scores derived from $\boldsymbol{A}^h$ may contain outliers with excessively high values, resulting in unstable estimation of importance scores. To mitigate this influence, we follow the prior work [3] and apply a max-pooling operation to these per-token importance scores over a sliding window of size $2W$ across recent tokens. Specifically, we denote $A_{j,i}^h$ as the attention score from the $j$-th query to the $i$-th key in $\boldsymbol{A}^h$. We obtain the stabilized

attention score $\tilde{A}^h$ by computing its $(i, j)$ entry, and finally obtain the importance score of retaining the $i$-th token in the KV cache as $I_i^h$ for each attention head $h$, as shown below:

$$\tilde{A}_{j,i}^h = \max\left(A_{j,i-W}^h, \ldots, A_{j,i}^h, \ldots, A_{j,i+W-1}^h\right), \quad I_i^h = \frac{1}{\alpha} \sum_{j=0}^{\alpha-1} \tilde{A}_{j,i}^h \in \mathbb{R}. \tag{4}$$

### 3.3 Redundancy Estimation via Semantic Similarity

To identify redundant tokens, we measure the semantic similarity between key states using cosine similarity. Tokens with high similarity to others are considered potentially redundant and can be selectively removed to optimize KV cache memory.

**Cosine Similarity between Key Tokens:** Given the key tokens $\boldsymbol{K}^h \in \mathbb{R}^{n \times d}$ for a specific head $h$, We first normalize each key vector $\boldsymbol{K}_i^h, \forall i \in [0, 1)$ into $\overline{\boldsymbol{K}}_i^h$, and then compute the cosine similarity matrix $\boldsymbol{S}^h$ using the normalized key vectors.

$$\overline{\boldsymbol{K}}_i^h = \frac{\boldsymbol{K}_i^h}{\|\boldsymbol{K}_i^h\|_2 + \epsilon} \in \mathbb{R}^d, \quad \boldsymbol{S}^h = \overline{\boldsymbol{K}}^h(\overline{\boldsymbol{K}}^h)^\top \in \mathbb{R}^{n \times n}, \quad S_{i,i}^h \leftarrow 0, \forall i \in [0, n), \tag{5}$$

where $\|\cdot\|_2$ is the L2 norm and $\epsilon$ is a small constant (e.g., $10^{-8}$) for numerical stability. To prevent tokens from being marked as redundant with themselves, we zero out the diagonal elements $S_{i,i}^h$.

**Enforce Retention of Recent Tokens.** While redundant, such tokens may still carry meaningful information. Thus, naively removing all redundant tokens can impair model performance. To address this, we retain only the $\beta$ most recently generated tokens among those exhibiting high similarity, as these later tokens tend to better support the model's decoding than earlier ones. To enforce this, we further zero out the similarity scores in $\boldsymbol{S}^h$ corresponding to these $\beta$ most recent similar tokens. Formally, for each token $i \in [0, n)$, we identify the set of indices of highly similar tokens: $\mathcal{I}_i^h = \{j \mid S_{j,i}^h > T, j \in [0, n)\}$, where $T$ is a fixed hyperparameter for similarity threshold. For this set, we extract the subject $\mathcal{I}_{i,\beta}^h \subseteq \mathcal{I}_i^h$, containing up to the $\beta$ largest indices—i.e., the $\beta$ most recent similar tokens to token $i$, or fewer if not enough such tokens exist. We then suppress their influence by zeroing out their similarity scores with token $i$ in $\boldsymbol{S}^h$, i.e., $S_{j,i}^h \leftarrow 0, \ \forall j \in \mathcal{I}_{i,\beta}^h$. This modification effectively nullifies the direct similarity links from token $i$ to its $\beta$ most recent highly similar tokens.

**Redundancy Score Estimation:** Finally, we compute normalized redundancy scores for all key tokens in Eq. (6). First, for each key token $i \in [0, n)$ in each head $h$, we compute its average similarity score $\bar{S}_i^h$. Intuitively, $\bar{S}_i^h$ measures how similar token i is, on average, to all other key tokens in the sequence. A high $\bar{S}_i^h$ indicates that the semantic content of token i is largely shared with other tokens, suggesting potential redundancy. Next, to obtain per-token redundancy scores $R_i^h$ within a fixed numerical range for each head $h$, we normalize $\bar{S}_i^h$ using a softmax operation. The resulting score $R_i^h$ reflects the redundancy of token $i$ for head $h$, with higher values indicating greater redundancy.

$$\bar{S}_i^h = \frac{1}{n} \sum_{j=0}^{n-1} S_{j,i}^h, \quad R_i^h = \left(\text{softmax}\left([\bar{S}_0^h, \ldots, \bar{S}_{n-1}^h]\right)\right)_i \tag{6}$$

### 3.4 Joint Selection Strategy for KV Cache Retention

To efficiently manage KV cache storage while retaining essential context, we employ a joint selection strategy that integrates both importance and redundancy scores. Given a predefined token budget $B_{budget}$ per attention head, our goal is to retain tokens that maximize information diversity while minimizing redundancy. The final selection score $Z_i^h$ for each token $i$ in head $h$ is computed as:

$$Z_i^h = \lambda I_i^h - (1 - \lambda) R_i^h, \tag{7}$$

where the importance score $I_i^h$ and the redundancy score $R_i^h$ are computed in Eq. (4) and Eq. (6) respectively. A higher $I_i^h$ indicates that a token is more important and should ideally be retained, while a higher $R_i^h$ suggests higher token redundancy. The hyperparameter $\lambda$ controls the trade-off

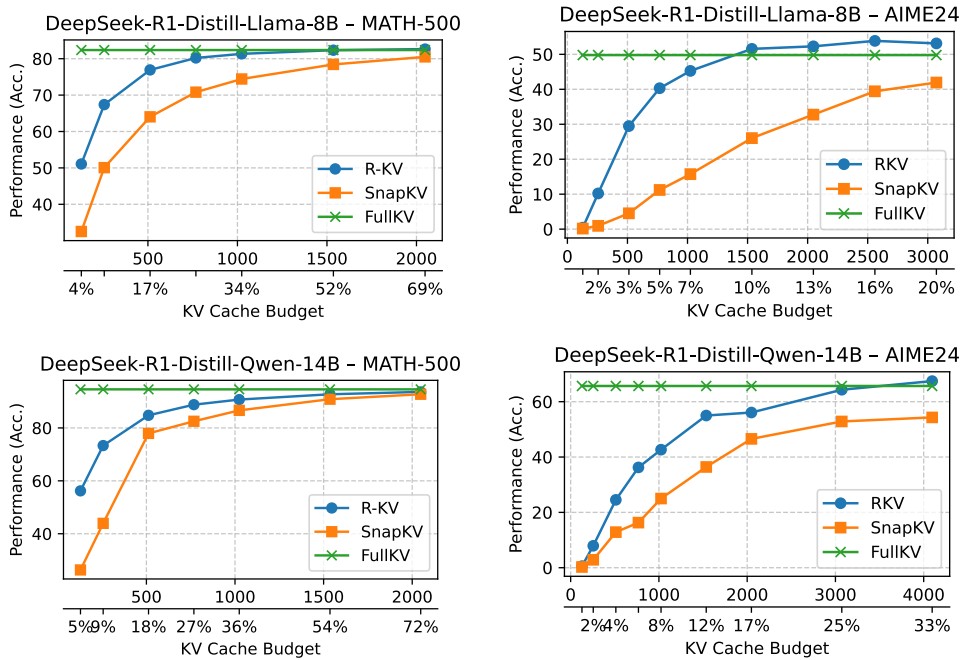

Figure 4: Results of R-KV compared with SnapKV and FullKV on the MATH-500 and AIME24 datasets for R1-Llama-8B (**top**) and R1-Qwen-14B (**bottom**). Results are reported as pass@1 based on 64 generated responses per question.

between prioritizing important tokens and reducing redundant tokens. We discuss the rationale for choosing $\lambda$ through a sensitivity analysis in §5.1. This strategy ensures that the KV cache prioritizes storing tokens that are both important and semantically diverse, thereby improving memory efficiency without compromising model performance.

# 4 Experiment

## 4.1 Experimental Setup

**Models and Datasets**    In our experiments, we use variants of the DeepSeek-R1 distilled model: DeepSeek-R1-Distill-Llama-8B, and DeepSeek-R1-Distill-Qwen-14B [1], which we refer to as R1-Llama-8B and R1-Qwen-14B, respectively, for brevity throughout the paper.

We evaluate the models' mathematical reasoning capabilities using three benchmarks: MATH-500 [8] and AIME 2024 [9].

**Hyperparameters**    We set $B_{\text{buffer}} = 128$, $\alpha = 8$ and $\lambda = 0.1$, with an analysis of $\lambda$ in §5.1.

**Baselines**    We compare our method against SnapKV [3], originally designed for long prefilling. To adapt it for decoding, we apply the same compression interval as our method, i.e., compressing the KV cache every 128 decoding steps using identical $B_{\text{budget}}$ and $B_{\text{buffer}}$. Our approach focuses on improving KV cache eviction through a hybrid strategy, and we therefore restrict comparison to state-of-the-art attention-based eviction methods. Budget allocation techniques (e.g., head-level [6] and layer-level [5]) are orthogonal to our work and not included. We also report results for FullKV, which retains the full KV cache and serves as the gold standard for decoding quality.

**Evaluation Setup**    We set the maximum generation length to 16,384 tokens for MATH-500 and 32,768 tokens for AIME 2024 and AIME 2025, because further increasing the generation length has shown no improvement on model performance on these datasets from our attempts. We find that using greedy decoding to evaluate long-output reasoning models results in significant variability across different setups. Following existing works [1], we utilize pass@$k$ evaluation [10] and report

pass@1 using a non-zero temperature. We use the recommended sampling temperature and top-$p$ value for each model, i.e., sampling temperature of $0.6$ and a top-$p$ value of $0.95$ for DeepSeek-R1 Distilled models. We generate $64$ responses for each question. Pass@1 is then calculated as Pass@1 $= \frac{1}{k}\sum_{i=1}^{k} p_i$, where $p_i$ denotes the correctness of the $i$-th response. This method provides more reliable performance estimates.

## 4.2 Results

The accuracy performance of R-KV compared with all baselines is shown in Figure 4, with detailed accuracy numbers in Appendix B.2. The KV cache budget ratio is calculated based on the KV cache budget and the average generation length of tokens, i.e., R1-Llama-8B: $2,979.1$ on MATH-500 and $15,535.8$ on AIME24; R1-Qwen-14B: $2,833.04$ on MATH-500 and $12,402$ on AIME24. Our method significantly outperforms the baseline SnapKV, achieving up to $40\%$ Acc. improvement. We provide two KV cache budget and performance analysis. Fixed budget analysis is more practical because when the model outputs longer (i.e., from $2,979.1$ on MATH-500 to $15,535.8$ on AIME24), the KV cache budget needed for lossless compression increases less (i.e., $512$). In the KV cache budget ratio perspective, the changes of lossless compression ratio is dominated by generation length.

**Ratio Budget**   For R1-Llama-8B, R-KV achieves lossless compression with $34\%$ KV cache budget on the MATH-500 dataset and with $10\%$ KV cache budget on the AIME-2024 dataset. Given $16\%$ KV cache budget, our method even surpasses the FullKV baseline, reaching $105\%$ of its accuracy. Similarly, for R1-Qwen-14B, R-KV achieves lossless compression with $54\%$ KV cache budget on the MATH-500 dataset and with $25\%$ KV cache budget on the AIME-2024 dataset. Given $33\%$ KV cache budget, our method achieves $105\%$ of FullKV accuracy.

**Fixed Budget**   For R1-Llama-8B, R-KV achieves lossless compression with $1024$ KV cache budget on the MATH-500 dataset and with $1536$ KV cache budget on the AIME-2024 dataset. For R1-Llama-8B, R-KV achieves lossless compression with $1536$ KV cache budget on the MATH-500 dataset and with $3072$ KV cache budget on AIME-2024.

## 5 Discussion

### 5.1 How to Choose $\lambda$?

Figure 5 shows the distributions of the Importance Score ($I^h$) and Redundancy Estimation ($R^h$) for head $h = 0$ at the top layer ($N_{\text{layer}} = 31$). The figure reveals that $I^h$ is sparse and dominated by a few outlier values, while the similarity distributions (which inform $R^h$) are relatively dense. When $\lambda = 0$, the token retention strategy is overned entirely by Redundancy Estimation ($R^h$). As shown in Figure 5, the initial four tokens are not guaranteed to be preserved. As highlighted by prior work [7], evicting these initial tokens can severely impair the generative capabilities of LLMs. Therefore, it is crucial to select a $\lambda$ value that starts from at least $0.01$. On the other hand, as $\lambda$ increases beyond $0.1$, the selection metric becomes increasingly dominated by attention scores. These observations suggest that an optimal $\lambda$ lies within the range of $0.01 \leq \lambda \leq 0.1$, effectively balancing the contributions of Importance Score and Redundancy Estimation.

Figure 6 presents the accuracy (Acc.) performance of R-KV on the DeepSeek-Distill-R1-Llama-8B model using the MATH-500 dataset. The results further guide the choice of $\lambda$ for optimal performance. The figure demonstrates that $\lambda = 0.1$ yields the highest accuracy. In contrast, strategies relying solely on redundancy ($\lambda = 0$) or solely on attention ($\lambda = 1$) exhibit the poorest performance, underscoring the complementary nature of these two metrics and the importance of a balanced approach. Thus, based on this finding, we select $\alpha = 0.1$ for all evaluations detailed in Figure 4.

### 5.2 Failure of Attention-Based Methods to Capture Redundancy

To thoroughly investigate the advantages of R-KV's hybrid selection metrics (combining attention and redundancy) over pure attention-based importance metrics, we compared the tokens selected by R-KV against those chosen by a pure attention-based method (SnapKV). We present a case where R-KV correctly completes the task while the comparison method fails. As illustrated in  Figure 7,

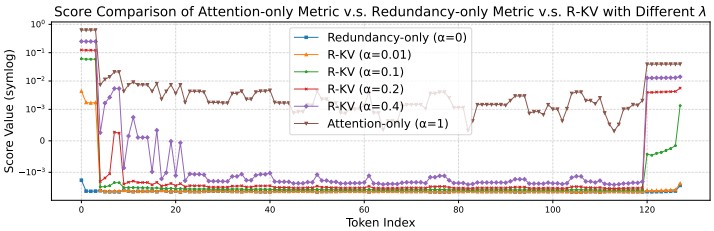
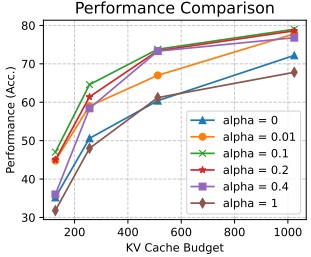

Figure 5: KV sekection score comparison of attention-only metric v.s. redundancy-only metric v.s. R-KV with different $\lambda$. When $\lambda \geq 0.1$, the selection score starts to be dominated by attention score.

Figure 6: Performance Comparison of the same methods as Figure 5.

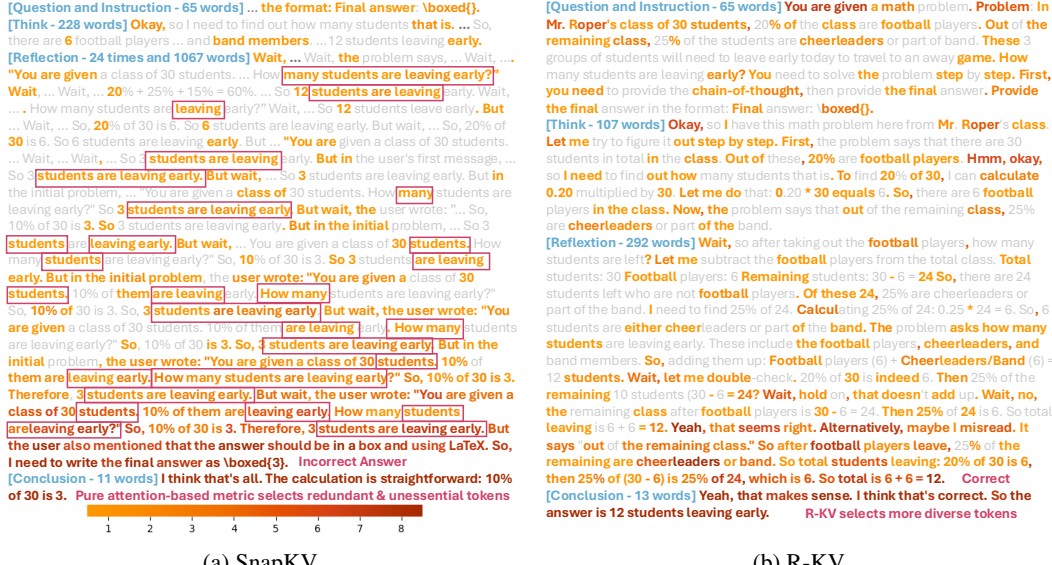

(a) SnapKV

(b) R-KV

Figure 7: Comparison of selected key-value (KV) tokens for an example between SnapKV (left) and R-KV (right). Grey tokens are unselected, while the gradient from light to dark red indicates the number of attention heads selecting each token (darker = more heads). R-KV selects a more diverse and broadly distributed set of tokens, capturing richer contextual information.

grey tokens represent unselected tokens, while the gradient from light orange to red indicates the number of heads selecting each token, with darker red signifying selection by more heads.

When considering the tokens selected by all heads, we observe that R-KV selects a more diverse set of tokens that cover a broader range and contain more effective information. These selections are more evenly distributed throughout the decoded output, capturing a more comprehensive context representation. In contrast, SnapKV's selected tokens exhibit more limited coverage. It tends to favor tokens positioned close to the query token, which are often selected multiple times by various heads, indicating a concentration of attention in localized areas. Furthermore, SnapKV also selects tokens that are not in close proximity to the query but still constitute largely redundant and unimportant segments (i.e., "3 students are leaving early." and "But in the initial").

## 5.3 Efficiency Analysis

**Memory Saving** R-KV achieves improved memory efficiency by allocating fixed-size buffers for both the retained KV cache and newly generated tokens. Unlike FullKV, which scales memory linearly with sequence length, R-KV 's memory footprint remains constant, enabling substantial savings during long-form generation. Detailed memory accounting is provided in Appendix C.1.

| Gen. Length | Method | Budget | Mem. Saving (%) | Batch | Throughput (tok/s) | Tokens Gen. | Dec. Time (s) |
|---|---|---|---|---|---|---|---|
| 8K | FullKV | – | – | 1 | 75.44 | 8 094 | 107.30 |
| | | – | – | 62 (max) | 849.13 | 501 828 | 590.99 |
| | R-KV | Fixed – 1024 | 87.50 | 1 | 80.46 | 8 094 | 100.60 |
| | | Fixed – 1024 | 87.50 | 402 (max) | 3 251.52 | 3 253 788 | 1 000.70 |
| | | Fixed – 1536 | 81.25 | 287 (max) | 2 525.75 | 6 546 972 | 919.72 |
| | | Ratio – 10% – 819 | 90.00 | 479 (max) | 3 809.15 | 3 877 026 | 1 017.82 |
| | | Ratio – 34% – 2 785 | 66.00 | 167 (max) | 1 608.01 | 1 351 698 | 840.61 |
| | | Ratio – 54% – 4 423 | 46.00 | 105 (max) | 1 257.83 | 849 870 | 675.66 |
| 16K | FullKV | – | – | 1 | 69.41 | 16 286 | 234.65 |
| | | – | – | 30 (max) | 347.03 | 488 580 | 1 407.89 |
| | R-KV | Fixed – 1024 | 93.75 | 1 | 80.95 | 16 286 | 201.18 |
| | | Fixed – 1024 | 93.75 | 402 (max) | 3 188.82 | 6 546 972 | 2 053.10 |
| | | Fixed – 1536 | 90.63 | 287 (max) | 2 447.61 | 4 674 082 | 1 909.65 |
| | | Ratio – 10% – 1 638 | 90.00 | 271 (max) | 2 300.28 | 4 413 506 | 1 918.68 |
| | | Ratio – 34% – 5 570 | 66.00 | 82 (max) | 797.43 | 1 335 452 | 1 674.70 |
| | | Ratio – 54% – 8 847 | 46.00 | 46 (max) | 584.77 | 749 156 | 1 281.12 |

Table 1: Memory saving, throughput, and decoding-time comparison for Llama3-8B under various generation length and KV cache compression budget settings.

**Computation Overhead**   While R-KV introduces additional computation for importance and redundancy scoring, the total overhead is modest and often outweighed by the reduced attention cost over a compressed KV cache. This trade-off becomes increasingly favorable as sequence length grows. Complexity comparisons can be found in Appendix C.1

**Real-time analysis**   We present the real-time analysis of memory saving and end-to-end throughput improvement in Table 1. When the batch size is 1, R-KV exhibits a slight throughput advantage over FullKV. This suggests that the acceleration achieved by R-KV through reduced attention computation outweighs computational overhead of R-KV. However, this direct speedup constitutes a minor portion of the overall benefit. The primary throughput improvement from R-KV stems from enabling significantly larger inference batch sizes due to KV cache compression.

We evaluate end-to-end throughput under both ratio-based and fixed KV cache budgets. R-KV consistently enables much larger batch sizes and higher throughput than FullKV, with benefits becoming more pronounced at longer sequence lengths. For example, at a sequence length of 16K, R-KV achieves up to $9\times$ larger batch sizes and over $6.6\times$ higher throughput under a 10% compression ratio, and $13.4\times$ larger batch sizes with $9.2\times$ throughput under a fixed budget of 1024. Detailed analysis are provided in Appendix C.2.

# 6   Related Work

**KV Cache Compression**   The optimization of KV cache memory efficiency in LLMs has garnered increasing attention as model sizes and context windows expand. Existing approaches primarily fall into three categories: dynamic token eviction[3, 11, 12], quantization[13, 14, 15], merging[16, 17, 18], and low-rank decomposition[19, 20, 21]. Previous eviction methods like SnapKV[3], PyramidKV[5], Ada-KV[22], HeadKV[6] dynamically prune tokens based on attention scores, but mainly focus on evicting tokens for prefilling stage. StreamingLLM[7] and H2O[4] are proposed for decoding. However, these general-purpose techniques often struggle with reasoning-intensive tasks, where aggressive eviction risks disrupting critical intermediate steps in CoT, and suffers from reasoning models' inherent redundancy.

**Efficient Reasoning**   Recent works in efficient reasoning focus on training the model to generate less CoT without sacrificing performance. [23, 24, 25] use RL optimization with length penalty rewards to encourage models to produce more concise chains-of-thought (CoT). [26, 27] employs variable-length CoT datasets to supervised fine-tune (SFT) the LLM to reduce token usage while preserving reasoning correctness. Both RL and SFT methods require additional training. [27, 28, 29] use test-time prompting to reduce generation length, but these methods may hurt the performance. As a KV cache compression work for reasoning models, R-KV is able to achieve lossless compression without extensive training and prompting.

# 7  Conclusion

We introduced R-KV, a novel decoding-time KV cache compression method tailored to the challenges of complex reasoning in large language models (LLMs). Reasoning models often generate long, redundant outputs that impose substantial memory and computational burdens during inference. R-KV addresses this by jointly scoring token importance and redundancy, enabling the retention of essential reasoning content while discarding repetitive or uninformative tokens. This dynamic and attention-guided strategy allows R-KV to preserve nearly full model performance using only 10–34% of the original KV cache—substantially outperforming prior compression methods.

Extensive throughput and efficiency analysis demonstrate that R-KV enables up to 13× larger batch sizes and over 9× speedup in long-sequence generation scenarios compared to FullKV, with particularly strong gains under constrained memory budgets. With its training-free and model-agnostic design, R-KV provides a scalable and deployment-ready solution for reasoning LLMs, especially in streamlining the rollout phase of reinforcement learning workflows.

# 8  Acknowledgement

Research reported in this publication was partially supported by the National Science Foundation under Award Number IIS-2449768. The content is solely the responsibility of the authors and does not necessarily represent the official views of the National Science Foundation.

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

# A  Method

## A.1  Algorithm

The pseudo-code of the method is shown in Algorithm 1.

## A.2  Implementation Details

**Max Pooling of Attention Weights**  Latest open-source LLMs [30, 31] have widely adopted Grouped-Query Attention (GQA) [32], where multiple query heads share a common pair of key-value heads to substantially reduce memory access overhead during inference. In key-value (KV) cache eviction strategies, it's thus often necessary to downscale attention scores from (Q_head, seq_len, seq_len) to (KV_head, seq_len, seq_len). While previous works such as SnapKV [3] have predominantly employed mean pooling to aggregate attention scores across query head groups, we hypothesize that max pooling could better preserve the most critical tokens for each query head. Our empirical results demonstrate that max pooling leads to improved performance, and we adopt it for all main experiments.

**Calibration of SnapKV's Observation Window Mask**  The official implementation of SnapKV applied an upper triangular attention mask to the attention weights matrix to enforce causality. The attention weights matrix is then processed with softmax, slicing, and summation to obtain observation window scores for each prefix token.

They adopted an upper triangle to prevent tokens in the observation window from seeing tokens after them, and then applied softmax and summation. However, their implementation does not account for the fact that tokens within the observation window absorb part of the attention weight originally assigned to prefix tokens, thereby disrupting the normalization property.

In our implementation, we address this issue by first slicing the attention weights before applying softmax. This approach ensures proper normalization and leads to significantly better scores in our tests.

```
1  if not fix_obs_window:
2      mask = torch.full(
```

**Algorithm 1** R-KV: $Q_{\text{obs}}$ are query states for $\alpha$ observation tokens, $K_{\text{full}}, V_{\text{full}}$ are the full KV cache states of length $L_{\text{full}}$.

1: **procedure** R-KV($(\boldsymbol{K}_{\text{full}}, \boldsymbol{V}_{\text{full}}), L_{\text{full}}, L_{\text{budget}}, Q_{\text{obs}}, \alpha, B_{\text{budget}}, B_{\text{buffer}}, T, \beta, \lambda, \epsilon, H, d_k$)
2:     **if** $L_{\text{full}} - L_{\text{budget}} < B_{\text{buffer}}$ **then**                 ▷ Check if compression is triggered
3:         **return** $(\boldsymbol{K}_{\text{full}}, \boldsymbol{V}_{\text{full}})$
4:     **end if**
5:     $(\boldsymbol{K}_{\text{obs}}, \boldsymbol{V}_{\text{obs}}) \leftarrow$ last $\alpha$ tokens of $(\boldsymbol{K}_{\text{full}}, \boldsymbol{V}_{\text{full}})$
6:     $(\boldsymbol{K}_{\text{cand}}, \boldsymbol{V}_{\text{cand}}) \leftarrow$ first $(L_{\text{full}} - \alpha)$ tokens of $(\boldsymbol{K}_{\text{full}}, \boldsymbol{V}_{\text{full}})$
7:     $N_c \leftarrow L_{\text{full}} - \alpha$                            ▷ Number of candidate tokens
8:     **if** $N_c \leq B_{\text{budget}}$ **then**
9:         **return** $(\boldsymbol{K}_{\text{full}}, \boldsymbol{V}_{\text{full}})$          ▷ Not enough candidates to prune beyond budget
10:     **end if**
11:     **for** each head $h = 0 \ldots H - 1$ **do**
12:         Compute attention matrix $\boldsymbol{A}^h \in \mathbb{R}^{\alpha \times N_c}$ using $Q_{\text{obs}}^h$ and $\boldsymbol{K}_{\text{cand}}^h$ ▷ Handles MHA/GQA as per Eqs. (1)-(3) from text
13:         **for** $k = 0 \ldots N_c - 1$ **do**                ▷ For each candidate token $k$
14:             $I'_{k,h} \leftarrow \frac{1}{\alpha} \sum_{q=0}^{\alpha-1} (\boldsymbol{A}^h)_{qk}$       ▷ $q$: observation token, $k$: candidate token
15:         **end for**
16:         $\{I_{k,h}\}_{k=0}^{N_c-1} \leftarrow$ 1D-Pooling$(\{I'_{k,h}\}_{k=0}^{N_c-1})$
17:     **end for**
18:     **for** each head $h = 0 \ldots H - 1$ **do**
19:         $\boldsymbol{K}_{\text{norm}}^h \in \mathbb{R}^{N_c \times d_k}$; For $k = 0 \ldots N_c - 1$, $\boldsymbol{K}_{\text{norm},k}^h \leftarrow \boldsymbol{K}_{cand,k}^h / (\|\boldsymbol{K}_{cand,k}^h\|_2 + \epsilon)$
20:         $\boldsymbol{S}^h \leftarrow \boldsymbol{K}_{\text{norm}}^h (\boldsymbol{K}_{\text{norm}}^h)^\top$    ▷ Cosine Similarity Matrix Computation, similarity matrix $\boldsymbol{S}^h \in \mathbb{R}^{N_c \times N_c}$
21:         **for** $k = 0 \ldots N_c - 1$ **do**                   ▷ Prevent Self-Redundancy
22:             $(\boldsymbol{S}^h)_{kk} \leftarrow 0$
23:         **end for**
24:         $B_{uv}^h \leftarrow ((\boldsymbol{S}^h)_{uv} > T?1 : 0)$ for $u, v \in \{0, \ldots, N_c - 1\}$ ▷ Identify Highly Similar Pairs
25:         **for** $u = 0 \ldots N_c - 1$ **do**                  ▷ Enforce Retention of Recent Tokens
26:             $T_u^h \leftarrow \{v \mid B_{uv}^h = 1, v \in \{0, \ldots, N_c - 1\}\}$
27:             $T_{u,\beta}^h \leftarrow$ subset of $T_u^h$ with up to $\beta$ largest indices $v$.
28:             **for** $v' \in T_{u,\beta}^h$ **do**
29:                 $(\boldsymbol{S}^h)_{u,v'} \leftarrow 0$
30:             **end for**                       ▷ $\boldsymbol{S}^h$ is now modified
31:         **end for**
32:         Let $\bar{\mathbf{S}}^h \in \mathbb{R}^{N_c}$ where $(\bar{\boldsymbol{S}}^h)_u \leftarrow \frac{1}{N_c} \sum_{v=0}^{N_c-1} (\boldsymbol{S}^h)_{uv}$
33:         **for** $u = 0 \ldots N_c - 1$ **do**
34:             $R_{u,h} \leftarrow (\text{softmax}(\bar{\mathbf{S}}^h))_u$
35:         **end for**
36:     **end for**
37:     **for** each head $h = 0 \ldots H - 1$ **do**
38:         **for** $k = 0 \ldots N_c - 1$ **do**
39:             $\text{Score}_{k,h} \leftarrow \lambda I_{k,h} - (1 - \lambda) R_{k,h}$
40:         **end for**
41:     **end for**
42:     Let $\text{AggScore} \in \mathbb{R}^{N_c}$
43:     **for** $k = 0 \ldots N_c - 1$ **do**
44:         $\text{AggScore}_k \leftarrow \text{mean}_h(\text{Score}_{k,h})$          ▷ Aggregate scores across heads
45:     **end for**
46:     $Idx_{sel} \leftarrow$ indices of top-$B_{budget}$ tokens from $\{0, \ldots, N_c - 1\}$ based on AggScore
47:     $\boldsymbol{K}_{cand\_sel} \leftarrow \boldsymbol{K}_{cand}[Idx_{sel}]; \boldsymbol{V}_{cand\_sel} \leftarrow \boldsymbol{V}_{cand}[Idx_{sel}]$
48:     $\boldsymbol{K}_{comp} \leftarrow \text{concatenate}(\boldsymbol{K}_{cand\_sel}, \boldsymbol{K}_{obs})$              ▷ Order might vary
49:     $\boldsymbol{V}_{comp} \leftarrow \text{concatenate}(\boldsymbol{V}_{cand\_sel}, \boldsymbol{V}_{obs})$
50:     $L_{prev\_comp} \leftarrow B_{budget} + \alpha$                ▷ Update length for next cycle
51:     **return** $(\boldsymbol{K}_{comp}, \boldsymbol{V}_{comp})$
52: **end procedure**

| Model | Benchmark | Method | 128 | 256 | 512 | 768 | 1 024 | 1 536 | 2 048 | 2 560 | 3 072 | 4 096 |
|---|---|---|---|---|---|---|---|---|---|---|---|---|
| Llama3-8B | MATH | FullKV | 82.38 | 82.38 | 82.38 | 82.38 | 82.38 | 82.38 | 82.38 | – | – | – |
| | | R-KV | 51.08 | 67.39 | 76.92 | 80.21 | 81.34 | 82.34 | 82.65 | – | – | – |
| | | SnapKV | 32.53 | 50.07 | 64.03 | 70.81 | 74.43 | 78.43 | 80.50 | – | – | – |
| | AIME24 | FullKV | 49.79 | 49.79 | 49.79 | 49.79 | 49.79 | 49.79 | 49.79 | 49.79 | 49.79 | – |
| | | R-KV | 0.42 | 10.21 | 29.48 | 40.31 | 45.26 | 51.56 | 52.29 | 53.85 | 53.13 | – |
| | | SnapKV | 0.16 | 0.94 | 4.53 | 11.20 | 15.73 | 26.04 | 32.76 | 39.43 | 41.93 | – |
| Qwen-14B | MATH | FullKV | 94.58 | 94.58 | 94.58 | 94.58 | 94.58 | 94.58 | 94.58 | – | – | – |
| | | R-KV | 56.21 | 73.33 | 84.77 | 88.79 | 90.72 | 92.72 | 93.62 | – | – | – |
| | | SnapKV | 26.32 | 43.93 | 77.93 | 82.52 | 86.63 | 90.86 | 92.73 | – | – | – |
| | AIME24 | FullKV | 65.68 | 65.68 | 65.68 | 65.68 | 65.68 | 65.68 | 65.68 | – | 65.68 | 65.68 |
| | | R-KV | 0.57 | 7.92 | 24.53 | 36.25 | 42.66 | 55.00 | 56.09 | – | 64.32 | 67.45 |
| | | SnapKV | 0.26 | 2.86 | 12.86 | 16.30 | 25.00 | 36.41 | 46.56 | – | 52.86 | 54.32 |

Table 2: Accuracy (%) of **Llama3-8B** and **Qwen-14B** on the MATH and AIME24 benchmarks under different memory-optimization methods across context lengths. "–" denotes configurations that were not evaluated.

```
3            (self.window_size, self.window_size),
4            torch.finfo(attn_weights.dtype).min,
5            device=attn_weights.device,
6        )
7        mask_cond = torch.arange(mask.size(-1), device=attn_weights.device)
8        mask.masked_fill_(mask_cond < (mask_cond + 1).view(mask.size(-1), 1), 0)
9        mask = mask.to(attn_weights.device)
10        attention_mask = mask[None, None, :, :]
11        attn_weights[
12            :, :, -self.window_size :, -self.window_size :
13        ] += attention_mask
14
15        attn_weights = nn.functional.softmax(
16            attn_weights, dim=-1, dtype=torch.float32
17        ).to(query_states.dtype)
18
19        attn_weights_sum = attn_weights[
20            :, :, -self.window_size :, : -self.window_size
21        ].mean(dim=-2)
22    else:
23        attn_weights_sum = (
24            nn.functional.softmax(
25                attn_weights[:, :, -self.window_size :, : -self.window_size],
26                dim=-1,
27                dtype=torch.float32,
28            )
29            .mean(dim=-2)
30            .to(query_states.dtype)
31        )
```

# B   Experiment

## B.1   Devices

We use NVIDIA A100 80G to finish all the experiments.

## B.2   Main Results

See Table 2.

# C Efficiency

## C.1 Complexity Analysis of Memory and Computation

**Memory Saving**  As discussed in §3.1, we need to allocate memory for the KV cache budget $M_{\text{budget}} \in \mathbb{R}^{b \times B_{\text{budget}} \times N_{\text{layer}} \times N_{\text{head}} \times d}$ to retain $B_{\text{budget}}$ KV cache tokens, and for the buffer $M_{\text{buffer}} \in \mathbb{R}^{b \times B_{\text{buffer}} \times N_{\text{layer}} \times N_{\text{head}} \times d}$ to store $B_{\text{buffer}}$ newly generated KV cache tokens during the generation of a text segment. Here, $b$ is the batch size, $N_{\text{layer}}$ is the number of Transformer layers, $N_{\text{head}}$ is the number of attention heads, and $d$ is the dimension of attention heads. In addition, we also need to allocate memory for the model weight $M_\theta$. During decoding, the previous query states are typically discarded by default, so we use a query cache to store the last $\alpha$ tokens in the query state, consuming memory of $M_\alpha \in \mathbb{R}^{b \times \alpha \times N_{\text{layer}} \times N_{\text{head}} \times d}$. In summary, R-KV requires memory of $M_{\text{total}} = M_\theta + M_{\text{budget}} + M_{\text{buffer}} + M_\alpha$ during generation. In comparison to FullKV without KV cache compression, generating $B_{\text{full}}$ tokens requires memory of $M_{\text{full}} \in \mathbb{R}^{b \times B_{\text{full}} \times N_{\text{layer}} \times N_{\text{head}} \times D_{\text{head}}}$ to retain $B_{\text{full}}$ KV tokens, and memory of the model weight $M_0$. Therefore, the memory saved by our method w.r.t. FullKV is: $M_{\text{saving}} = M_{\text{full}} - M_{\text{budget}} - M_{\text{buffer}} - M_\alpha$.

**Computation Overhead**  The computational complexity of importance scoring (See §3.2) is $O(\alpha B_{\text{budget}})$ while redundancy estimation (see §3.3) has complexity $O(B_{\text{budget}}^2)$. Thus, the total overhead incurred during each generation segment is $O(\alpha B_{\text{budget}} + B_{\text{budget}}^2)$. The generation complexity without KV cache compression is $O(B_{\text{full}} B_{\text{buffer}})$, whereas the complexity with KV cache compression is $O((B_{\text{budget}} + B_{\text{buffer}}) B_{\text{buffer}})$. For reasoning models, $B_{\text{full}}$ tends to be large because of the long generation length, and using a relatively small $B_{\text{budget}}$ value can efficiently reduce computation cost. The effectiveness of this approach depends on depends on whether the speedup gained by attending over a reduced KV cache outweighs the overhead of computing the compression scores—i.e., the combined cost of importance and redundancy scores, $(O(\alpha B_{\text{budget}}) + O(B_{\text{budget}}^2))$.

## C.2 Detailed Analysis of Throughput Results

We analyze the end-to-end throughput from two perspectives: ratio budget and fixed budget.

**Ratio Budget:** section 4.2 indicates that for DeepSeek-R1-Distill-Llama-8B, lossless compression (i.e., model performance equivalent to no KV compression) is achievable when the KV budget ratio, relative to the output length, is between 10% and 34%. For DeepSeek-R1-Distill-Qwen-14B, this range for lossless compression is 25% to 54% of the output length. Consequentlywe investigated the maximum achievable batch size and corresponding throughput for R-KV at compression ratios of 10%, 34%, and 54%, comparing these against the maximum batch size and throughput of FullKV using DeepSeek-R1-Distill-Llama-8B. In 8K sequence length setting, at a 54% compression ratio, R-KV allows for a batch size 1.7 × larger than FullKV, resulting in 1.5 × the throughput. At a 10% compression ratio, R-KV achieves a 7.7 × increase in batch size and a 4.5 × increase in throughput compared to FullKV. For a 16K sequence length setting, at 54% compression, the batch size is 1.5 × that of FullKV, and the throughput is 1.7 × higher. At 10% compression, R-KV supports a 9 × larger batch size, delivering 6.6 × the throughput. We observe that for smaller batch sizes (e.g., less than 128), throughput scales nearly linearly with increasing batch size. However, for larger batch sizes this linear scaling diminishes as inference on the NVIDIA A100 GPU becomes compute-bound.

**Fixed Budget:** We also conducted an analysis under a fixed KV cache budget. With an output length of 8K and a fixed budget $B_{\text{budget}} = 1024$, R-KV enables a batch size 6.48 × larger than FullKV, yielding 3.8 × the throughput. At $B_{\text{budget}} = 1536$, the batch size is 4.6 × larger, and throughput is 3 × that of FullKV. For an output length of 16K and $B_{\text{budget}} = 1024$, R-KV achieves a 13.4 × increase in batch size and a 9.19 × increase in throughput. With $B_{\text{budget}} = 1536$, the batch size is 9.6 × larger, and throughput is 7.1 × higher. In the fixed budget scenario, the advantage of R-KV becomes more pronounced with longer generation lengths. This is because the KV cache size for R-KV under a fixed budget does not increase with the sequence length, unlike FullKV where the memory footprint grows linearly with the generation length, thus more severely limiting its maximum batch size.

| Gen. Length | Method | Budget | Mem. Saving (%) | Batch | Throughput (tok/s) | Tokens Gen. | Dec. Time (s) |
|---|---|---|---|---|---|---|---|
| 8K | FullKV | – | – | 1 | 75.44 | 8 094 | 107.30 |
| | | – | – | 62 (max) | 849.13 | 501 828 | 590.99 |
| | SnapKV | Fixed – 1024 | 87.50 | 1 | 81.26 | 8 094 | 99.60 |
| | | Fixed – 1024 | 87.50 | 402 (max) | 3 253.93 | 3 253 788 | 999.96 |
| | | Fixed – 1536 | 81.25 | 287 (max) | 2 525.25 | 2 322 978 | 919.90 |
| | | Fixed – 3072 | 62.50 | 150 (max) | 1 527.67 | 1 214 100 | 794.74 |
| | | Ratio – 10% – 819 | 90.00 | 479 (max) | 3 808.81 | 3 877 026 | 1 017.91 |
| | | Ratio – 34% – 2 785 | 66.00 | 167 (max) | 1 625.46 | 1 351 698 | 831.58 |
| | | Ratio – 54% – 4 423 | 46.00 | 105 (max) | 1 269.68 | 849 870 | 669.36 |
| | R-KV | Fixed – 1024 | 87.50 | 1 | 80.46 | 8 094 | 100.60 |
| | | Fixed – 1024 | 87.50 | 402 (max) | 3 251.52 | 3 253 788 | 1 000.70 |
| | | Fixed – 1536 | 81.25 | 287 (max) | 2 525.75 | 6 546 972 | 919.72 |
| | | Fixed – 3072 | 62.50 | 150 (max) | 1 520.99 | 1 214 100 | 798.23 |
| | | Ratio – 10% – 819 | 90.00 | 479 (max) | 3 809.15 | 3 877 026 | 1 017.82 |
| | | Ratio – 34% – 2 785 | 66.00 | 167 (max) | 1 608.01 | 1 351 698 | 840.61 |
| | | Ratio – 54% – 4 423 | 46.00 | 105 (max) | 1 257.83 | 849 870 | 675.66 |
| 16K | FullKV | – | – | 1 | 69.41 | 16 286 | 234.65 |
| | | – | – | 30 (max) | 347.03 | 488 580 | 1 407.89 |
| | SnapKV | Fixed – 1024 | 87.50 | 1 | 81.03 | 16 286 | 200.99 |
| | | Fixed – 1024 | 87.50 | 402 (max) | 3 202.17 | 6 546 972 | 2 044.54 |
| | | Fixed – 1536 | 81.25 | 287 (max) | 2 449.02 | 4 674 082 | 1 908.56 |
| | | Fixed – 3072 | 81.25 | 150 (max) | 1 413.84 | 2 442 900 | 1 727.84 |
| | | Ratio – 10% – 1 638 | 90.00 | 271 (max) | 2 306.26 | 4 413 506 | 1 913.71 |
| | | Ratio – 34% – 5 570 | 66.00 | 82 (max) | 798.42 | 1 335 452 | 1 672.61 |
| | | Ratio – 54% – 8 847 | 46.00 | 46 (max) | 586.43 | 749 156 | 1 277.48 |
| | R-KV | Fixed – 1024 | 93.75 | 1 | 80.95 | 16 286 | 201.18 |
| | | Fixed – 1024 | 93.75 | 402 (max) | 3 188.82 | 6 546 972 | 2 053.10 |
| | | Fixed – 1536 | 90.63 | 287 (max) | 2 447.61 | 4 674 082 | 1 909.65 |
| | | Fixed – 3072 | 81.25 | 150 (max) | 1 406.28 | 2 442 900 | 1 737.13 |
| | | Ratio – 10% – 1 638 | 90.00 | 271 (max) | 2 300.28 | 4 413 506 | 1 918.68 |
| | | Ratio – 34% – 5 570 | 66.00 | 82 (max) | 797.43 | 1 335 452 | 1 674.70 |
| | | Ratio – 54% – 8 847 | 46.00 | 46 (max) | 584.77 | 749 156 | 1 281.12 |

Table 3: Memory-saving, throughput, and decoding-time comparison for LLAMA3-8B under various generation lengths and KV-cache compression budgets.

## C.3 Results

Full results could be found at Table 3. While R-KV incurs a minor computational overhead for redundancy estimation compared with SnapKV, this results in a throughput that is only slightly lower, with a negligible difference of less than 1%.

## D Limitations

One limitation of our proposed KV cache compression method is its current compatibility with certain advanced attention mechanisms, such as paged attention. Adapting our compression technique to seamlessly integrate with such mechanisms presents a non-trivial challenge and may require further investigation. Additionally, the implementation of KV cache compression within existing serving frameworks can encounter practical difficulties, particularly if these frameworks lack native support or flexible interfaces for KV cache compression. In serving frameworks that do not offer specialized KV cache compression interfaces, the performance benefits of our method might be less pronounced. Without such interfaces, implementing KV cache compression may necessitate reallocating memory to store the compressed KV cache and subsequently deallocating the memory used for the original, uncompressed cache. This process of memory reallocation can introduce significant overhead, potentially offsetting some of the acceleration gains. In contrast, serving frameworks equipped with dedicated KV compression interfaces can handle these operations much more efficiently, avoiding such costly memory management tasks.

