# OpenReview forum: "R-KV: Redundancy-aware KV Cache Compression for Reasoning Models"
_NeurIPS.cc/2025/Conference — NeurIPS 2025 poster_

### Official Review · Reviewer_Dks4 · 2025-07-01

**Clarity:** 2
**Significance:** 2
**Originality:** 3
**Rating:** 4
**Confidence:** 4

**Summary:**

This paper tackles the problem of excessive redundant information in the KV cache during long chain‑of‑thought reasoning in large language models by introducing R‑KV, a training‑free, decoding‑time plugin. R‑KV computes, for each cached token, an importance score based on attention weights and a redundancy score based on semantic similarity, then employs a tunable trade‑off coefficient to retain only the most informative and diverse tokens. As a result, it achieves reasoning accuracy comparable to—or even exceeding—that of the full KV cache using substantially less cache space, while significantly reducing memory footprint and accelerating inference.

**Questions:**

1. In Figure 5, we observe that the primary difference between the λ = 0.1 configuration and the Redundancy‑only variant lies in the preservation of Attention Sink tokens and those at the tail positions. Could one instead explicitly designate certain positional tokens to retain, while applying the redundancy metric exclusively to all other tokens?

2. Since different backbone models exhibit varying degrees of inherent redundancy across datasets, how does this variability influence the efficacy and gains of the proposed method? Are there any deeper insights into how a model’s intrinsic redundancy profile interacts with R‑KV’s compression strategy?

**Ethical Concerns:**

["NO or VERY MINOR ethics concerns only"]

**Final Justification:**

**Resolved Issue**
- The rebuttal convincingly strengthens the paper on generalization and usability while leaving only moderate, non-blocking gaps in hyper-parameter coverage and technique interplay.
- Given the novelty of the redundancy-aware perspective, the clear empirical benefits, and the now-broader evidence base, the merits outweigh the remaining weaknesses.

**Remaining Issues**
- Pending empirical validation of interoperability with quantization.

Balancing these factors, I raise my rating from 3 (weak reject) to **4 (borderline accept)**.

**Limitations:**

yes

**Quality:**

2

**Strengths And Weaknesses:**

**Strengths**

- **Novel redundancy‑aware view of KV eviction.**
By introducing a combined importance‑and‑redundancy scoring mechanism, the paper overcomes the limitation of prior methods that rely solely on attention weights and thus fail to eliminate repetitious self‑reflections in the KV cache. This new perspective directly targets semantic duplication and yields a more information‑dense cache
.

- **Comprehensive and transparent experimental analysis.**
The authors evaluate R‑KV across multiple layers, λ‑settings and cache budgets, and directly compare against both vanilla attention‑based pruning and recent compression baselines. Clear plots (e.g. accuracy vs. cache size) and ablations (importance only vs. redundancy only vs. joint scoring) convincingly demonstrate the complementary benefits of each component
.

- **Lightweight, training‑free deployment.**
R‑KV operates entirely at decoding time and requires neither re‑training nor architectural changes, making it immediately pluggable into existing Transformer‑based inference pipelines with minimal engineering effort and no model‑specific adjustments
.

**Weaknesses**

- **Limited domain generalization.**
Experiments are confined to mathematical reasoning benchmarks (MATH‑500, AIME‑24) and a single distilled model family (DeepSeek‑R1). The absence of evaluations on other reasoning tasks leaves open questions about R‑KV’s effectiveness beyond purely math domains.

- **Unexamined hyperparameter sensitivity.**
The method depends on several tunable factors—redundancy threshold  T, recent‑token retention β—yet the paper lacks a systematic study of how robust performance is to these settings or guidance for selecting them in new contexts.

- **Figure annotation errors.**
Some plot legends and symbols differ from those used in the text, and certain figures contain outright errors. For example, in Figure 1 the “12” below “decoding” should read “14,” and in Figures 5 and 6 the legend erroneously spells λ as α.

- **Compatibility with other techniques.**
While R‑KV offers substantial speed and memory gains, its interaction with other techniques (e.g. quantization approaches) is not discussed. Potential conflicts or synergies with these established techniques warrant further exploration.

---

> ### Author Rebuttal · Authors · 2025-07-31
>
> We thank the reviewer for the insightful feedback, and we would like to address a few concerns:
>
> > **Q1**: Limited domain generalization
>
> **A1**: To address generalization beyond the original evaluation scope, we conducted additional experiments on the GPQA benchmark, which emphasizes scientific knowledge-based question answering. R-KV substantially outperformed the baseline method. These results clearly indicate that R-KV consistently outperforms SnapKV across both scientific QA and mathematical reasoning tasks. We also evaluated R-KV on a mathematical reasoning benchmark using a different family of reasoning models—QwQ—and observed consistently strong performance. These results demonstrate that R-KV is robust and effective across diverse reasoning domains and model families. We will include these extended evaluations in the revised version of the paper.
>
> **Deepseek-Distilled-Llama-8B - GPQA**:
>
> | Method-Budget | Overall Accuracy | Physics | Chemistry | Biology |
> |-------|------------------|---------|-----------|--------|
> | rkv-128 | 3.03% | 4.65% | 1.08% | 5.26% |
> | snapkv-128 | 0.00% | 0.00% | 0.00% | 0.00% |
> | rkv-256 | 17.17% | 22.09% | 9.68% | 31.58% |
> | snapkv-256 | 0.00% | 0.00% | 0.00% | 0.00% |
> | rkv-512 | 23.23% | 33.72% | 8.60% | 47.37% |
> | snapkv-512 | 1.52% | 3.49% | 0.00% | 0.00% |
>
> **QwQ**:
>
> | - | MATH Pass@1 | - | AIME24 | - | AIME25 Pass@1 | - |
> |--------|------------------|--------------------|---------------------|-----------------------|---------------------|-----------------------|
> | budget | R-KV | SnapKV | R-KV | SnapKV | R-KV | SnapKV |
> | 128    | 0.4812           | 0.1522             | 0.0215              | 0.0021                | 0.0052              | 0.0011                |
> | 256    | 0.6476           | 0.334              | 0.0734              | 0.0063                | 0.0151              | 0.0031                |
> | 512    | 0.8031           | 0.5581             | 0.1479              | 0.0308                | 0.0802              | 0.0142                |
> | 768    | 0.8585           | 0.7476             | 0.2251              | 0.0604                | 0.162               | 0.0355                |
> | 1024   | 0.8897           | 0.8359             | 0.3062              | 0.1742                | 0.213               | 0.1088                |
> | 1536   | 0.9323           | 0.9032             | 0.4229              | 0.2833                | 0.2845              | 0.1891                |
> | 2048   | 0.9457           | 0.938              | 0.5354              | 0.4646                | 0.3396              | 0.2567                |
> | 2560   | 0.9559           | 0.9549             | 0.5729              | 0.5458                | 0.3718              | 0.3124                |
> | 3072   | 0.9623           | 0.9583             | 0.6014              | 0.5735                | 0.4033              | 0.3402                |
>
> > **Q2**: Unexamined hyperparameter sensitivity
>
> **A2**: We conducted additional experiments to examine the sensitivity of our method to the recent-token retention parameter β. Our findings indicate that performance is indeed sensitive to β, with the optimal setting typically around 12.5% of the KV budget. We will include a more comprehensive analysis of this and other key hyperparameters in the revised version, along with practical guidance for selecting them.
>
> | R-KV Budget | Window Size | Window/Budget (%) | Value |
> |-------------|-------------|--------------------|--------|
> | 128         | 8           | 6.3%               | 47.0   |
> | 128         | 16          | 12.5%              | 48.0   |
> | 128         | 32          | 25.0%              | 46.4   |
> | 128         | 64          | 50.0%              | 42.0   |
> | 256         | 16          | 6.3%               | 64.6   |
> | 256         | 32          | 12.5%              | 65.4   |
> | 256         | 64          | 25.0%              | 64.4   |
> | 256         | 128         | 50.0%              | 62.0   |
>
> > **Q3:** Figure annotation errors
>
> **A3**: We appreciate the reviewer pointing this out and will correct the annotation errors in the revised version.
>
> > **Q4**: Compatibility with other techniques
>
> **A4:** We will include experiments and analysis of R-KV's compatibility with other techniques including quantization in the revised version.
>
> > **Q5**: Design based on Figure 5
>
> **A5**: We agree that explicitly designating certain positional tokens to retain—such as attention sink tokens or tail-position tokens—while applying the redundancy metric to others could potentially yield good results for specific models. However, the optimal number and position of tokens to preserve may vary significantly across different models and tasks. Additionally, tokens with large attention scores can also emerge at intermediate positions, depending on the model and input characteristics. R-KV offers a flexible and elegant alternative by eliminating the need for manual specification of such positions. Instead, it adopts a unified strategy based on real-time analysis during inference that consistently applies across all tokens, thereby avoiding model-specific or position-specific heuristics. This generality enables R-KV to adapt more broadly and reduces the need for task-specific tuning.
>
> > **Q6**: How does the model's intrinsic redundancy influence R-KV's performance
>
> **A6**: To clarify, by models’ inherent redundancy across datasets, we refer to the degree of output redundancy exhibited by a model when applied to different datasets. In our observations, a reasoning-oriented model typically shows a consistently high level of redundancy across datasets, whereas non-reasoning models exhibit little to no such redundancy. This suggests that redundancy is more tightly tied to the model parameters, rather than the dataset alone. We have used both a reasoning model and a non-reasoning model on the Math dataset, evaluating performance under different values of the redundancy threshold λ. The results indicate that for non-reasoning models, introducing redundancy control yields minimal to no performance improvement. In contrast, the reasoning model benefits significantly from R-KV, showing large gains under the same settings. These findings suggest that R-KV is particularly effective when applied to models with higher intrinsic redundancy. This aligns with our hypothesis that the more redundant the model’s output is, the greater the compression and performance benefits R-KV can provide. We will include this discussion and the supporting results in the revised version.

---

> > ### Comment · Reviewer_Dks4 · 2025-08-06
> >
> > I have carefully reviewed the authors’ rebuttal. The authors have satisfactorily addressed the weaknesses and concerns raised in my initial review. I am raising my overall score to 4.

---

### Official Review · Reviewer_vf9G · 2025-07-02

**Clarity:** 3
**Significance:** 2
**Originality:** 3
**Rating:** 4
**Confidence:** 4

**Summary:**

The paper introduces R-KV(Redundancy-aware KV Cache Compression), a training-free KV cache compression method for reasoning models that produce long, redundant outputs. R-KV works by jointly considering token importance (via attention) and redundancy (via cosine similarity) to selectively prune the KV cache. Evaluated on two distilled models across two mathematical reasoning benchmarks, the method achieves performance comparable to the full KV cache with a significantly smaller memory footprint, leading to substantial memory savings and inference speedups.

**Questions:**

**Questions:**
1. Unexplained Performance Boost. The paper reports that R-KV can surpass the full-cache baseline but does not provide a deep analysis of why this regularization-like effect occurs.
2. Discrepancy in Baseline Performance. The reported baseline scores (FullKV) appear lower than
publicly available benchmarks. For example, the paper reports 82.38% for Llama3-8B on MATH-
500 and 65.68% for Qwen-14B on AIME24, while official or other reported scores are higher (e.g.,
~89.1% and ~69.7%, respectively). This gap raises questions about the experimental setup and
could overstate the relative gains of R-KV.

**Suggestions:**
1. Provide stronger evidence for R‑KV’s redundancy elimination. The manuscript currently includes only a single case study to illustrate how R‑KV removes redundancy in long-chain-of-thought (CoT) reasoning, which may invite skepticism in the absence of further examples. We recommend that the authors bolster their claims with more compelling statistical analyses—such as an n‑gram redundancy analysis similar to that shown in Figure 2—to quantitatively demonstrate the method’s effectiveness.
2. Typo. In Section 5.1 (line 231), the word "overned" appears to be a typo and should likely be "governed".
3. Figure Inconsistency. The diagram in Figure 1 states "Let λ = 0.5". However, the paper's
experimental setup establish λ = 0.1 as the default value. It is suggested to update the figure to
reflect λ = 0.1 for consistency with the main text.

**Ethical Concerns:**

["NO or VERY MINOR ethics concerns only"]

**Final Justification:**

- Strengths: Problem is important, solution is simple, training-free, and now shown to transfer across two additional settings with compelling memory-/speed-to-accuracy trade-offs.
- Weaknesses: Breadth of evaluation and analysis depth remain limited; baseline score gap introduces residual uncertainty about the true magnitude of gains.

Given the added evidence and clarifications, the paper’s contributions outweigh its shortcomings, but only narrowly. I therefore maintain a **borderline accept** recommendation. Strengthening the baseline reproduction and providing the promised redundancy analysis would move the work into clear-accept territory.

**Limitations:**

yes

**Quality:**

3

**Strengths And Weaknesses:**

**Strengths**
1. Clear Problem Definition and Motivation. The paper precisely identifies and motivates a critical
bottleneck in deploying reasoning models—the long, redundant outputs generated during the
decoding phase.
2. Strong Empirical Performance. The method demonstrates significant effectiveness, achieving
performance comparable to, and in some cases surpassing, the uncompressed baseline while
using only a small fraction of the KV cache.
3. Simple yet principled algorithm. The method is training-free and model-agnostic, making it an
easy-to-deploy solution for improving inference efficiency without retraining.

**Weaknesses**
1. Limited Scope of Models and Tasks. The evaluation is limited to two distilled models on math
tasks. Given the method is training-free and computationally light, its generalizability should be
validated on a broader range of architectures and reasoning domains to strengthen its claims.
2. Some parameter settings have not been specified in the paper. For example, neither β nor T is mentioned in the hyperparameter configuration.

---

> ### Author Rebuttal · Authors · 2025-07-31
>
> We thank the reviewer for the insightful feedback, and we would like to clarify a few points:
>
> > **Q1**: Limited Scope of Models and Tasks.
>
> **A1**: We conduct additional evaluations of R-KV on the GPQA benchmark, which targets scientific knowledge-based question answering. Despite the increased complexity of this domain, R-KV consistently outperforms the baseline method. Moreover, we extended our evaluation to a different family of reasoning models—QwQ—on mathematical reasoning benchmarks, further validating the robustness of R-KV across architectures. In all cases, R-KV demonstrates strong and reliable performance, underscoring its adaptability to diverse reasoning domains. The consistent superiority of R-KV across varied KV budgets and datasets further confirms its robustness. We will include these extended results in the revised version of the paper.
>
> **Deepseek-Distilled-Llama-8B - GPQA**:
>
> | Method-Budget | Overall Accuracy | Physics | Chemistry | Biology |
> |-------|------------------|---------|-----------|--------|
> | rkv-128 | 3.03% | 4.65% | 1.08% | 5.26% |
> | snapkv-128 | 0.00% | 0.00% | 0.00% | 0.00% |
> | rkv-256 | 17.17% | 22.09% | 9.68% | 31.58% |
> | snapkv-256 | 0.00% | 0.00% | 0.00% | 0.00% |
> | rkv-512 | 23.23% | 33.72% | 8.60% | 47.37% |
> | snapkv-512 | 1.52% | 3.49% | 0.00% | 0.00% |
>
> **QwQ**:
>
> | - | MATH Pass@1 | - | AIME24 | - | AIME25 Pass@1 | - |
> |--------|------------------|--------------------|---------------------|-----------------------|---------------------|-----------------------|
> | budget | R-KV | SnapKV | R-KV | SnapKV | R-KV | SnapKV |
> | 128    | 0.4812           | 0.1522             | 0.0215              | 0.0021                | 0.0052              | 0.0011                |
> | 256    | 0.6476           | 0.334              | 0.0734              | 0.0063                | 0.0151              | 0.0031                |
> | 512    | 0.8031           | 0.5581             | 0.1479              | 0.0308                | 0.0802              | 0.0142                |
> | 768    | 0.8585           | 0.7476             | 0.2251              | 0.0604                | 0.162               | 0.0355                |
> | 1024   | 0.8897           | 0.8359             | 0.3062              | 0.1742                | 0.213               | 0.1088                |
> | 1536   | 0.9323           | 0.9032             | 0.4229              | 0.2833                | 0.2845              | 0.1891                |
> | 2048   | 0.9457           | 0.938              | 0.5354              | 0.4646                | 0.3396              | 0.2567                |
> | 2560   | 0.9559           | 0.9549             | 0.5729              | 0.5458                | 0.3718              | 0.3124                |
> | 3072   | 0.9623           | 0.9583             | 0.6014              | 0.5735                | 0.4033              | 0.3402                |
>
> > **Q3**: Some parameter settings have not been specified in the paper.
>
> **A3**: We have used β = 8 and temperature T = 0.5 in all experiments. These settings will be explicitly stated in the revised version.
>
> > **Q4**: Discrepancy in Baseline Performance
>
> **A4**: In all evaluations (including FullKV), we followed the official reports by using the same sampling hyperparameters: temperature = 0.6 and top-p = 0.95, and reported the average pass@1 over 64 samples. However, we were unable to exactly reproduce the official results, likely due to unreported details such as prompt formatting, sampling seed control, or other subtle differences in the inference pipeline. Although we could not replicate the official results precisely, we emphasize that all comparisons in our work were made under a controlled and consistent inference setup. Importantly, all our experiments—including both R-KV and baseline comparisons—were conducted under the same hyperparameters and experiment setup, ensuring that the relative performance differences reported in our paper remain valid and meaningful.
>
> > **Q5**: Provide stronger evidence for R‑KV’s redundancy elimination
>
> **A5**: We agree that stronger empirical evidence would enhance the claims. In the revised version, we will include a detailed statistical analysis comparing the retained KV tokens between baseline methods and R-KV. Specifically, we will perform an n-gram redundancy analysis—similar to the one presented in Figure 2—to quantitatively evaluate how R-KV reduces repetitive information in the KV cache. This analysis will further substantiate the effectiveness of our redundancy-aware selection strategy.
>
> > **Q6 and Q7**: Typo and Figure Inconsistency
>
> **A6**: We acknowledge the inconsistencies and will ensure they are corrected in the revised manuscript to improve clarity and presentation quality.

---

> > ### Comment · Reviewer_vf9G · 2025-08-06
> > **Response to the rebuttal**
> >
> > Thank you again for the thorough follow-up experiments and clarifications. I appreciate the additional GPQA and QwQ results and the effort to specify β and T. After carefully reviewing the new evidence, I have raised the Quality score from 2 to 3 (good). All other scores remain unchanged.

---

### Official Review · Reviewer_cogD · 2025-07-03

**Clarity:** 3
**Significance:** 3
**Originality:** 2
**Rating:** 4
**Confidence:** 3

**Summary:**

This paper addresses a critical challenge in deploying large language models (LLMs) for reasoning tasks: the excessive GPU memory consumption caused by long chain-of-thought (CoT) outputs. These verbose generations often lead to large key-value (KV) caches during autoregressive decoding. While prior work has attempted to reduce memory usage by selecting important tokens based on attention scores, this paper claims that this is insufficient.

Their key insight is that even among high-attention tokens, there is substantial redundancy. Many tokens are semantically repetitive and contribute little to final task performance. Retaining such redundant tokens crowds out more diverse and informative context, ultimately hurting reasoning accuracy.

To address this, the paper proposes **R-KV**, a **redundancy-aware KV cache compression method** that considers both:

1. **Token importance**, estimated via attention scores;
2. **Token redundancy**, measured through semantic similarity (cosine similarity of key vectors);
3. A **joint selection strategy**, which balances these two aspects using a tunable parameter λ, to retain a subset of KV tokens that are both important and diverse.

R-KV operates during the decoding stage---where output grows rapidly---and performs cache compression at fixed intervals. It maintains two memory buffers: a cache of size $B_{\text{budget}}$ to store selected key-value tokens, and a buffer of size $B_{\text{buffer}}$ to temporarily hold newly generated tokens.

After generating each $B_{\text{buffer}}$ tokens, the algorithm first preserves the most recent $\alpha$ tokens (observation tokens) to ensure continuity. Then, from the combined pool of $B_{\text{budget}} + B_{\text{buffer}} - \alpha$ candidate tokens, it selects the top $B_{\text{budget}} - \alpha$ tokens according to the joint importance--redundancy score. These tokens, along with the $\alpha$ most recent ones, are retained for the next decoding step, effectively compressing the KV cache while preserving useful and diverse context.

Experiments are conducted on two mathematical reasoning benchmarks: **MATH-500** and **AIME 2024**, using **DeepSeek-R1-Distill-Llama-8B** and **Qwen-14B** models. Results show that:

- R-KV achieves near-parity or even **105% of full KV accuracy** using only **10–34%** of the original KV cache.
- It **outperforms the prior state-of-the-art method SnapKV by up to 40%** in accuracy.
- It enables **up to 90% memory savings**, **13× larger batch sizes**, and **9× higher throughput**, significantly improving inference efficiency under constrained memory settings..

Overall, R-KV is a training-free and model-agnostic method that demonstrates strong effectiveness in both accuracy and throughput.

**Questions:**

Questions are included in the weaknesses.

**Ethical Concerns:**

["NO or VERY MINOR ethics concerns only"]

**Final Justification:**

I have read the authors' response, and I keep my score unchanged. The method, which combines importance and redundancy scores for pruning KV states, is practically effective. I give a positive score because we need more research into understanding which parts of the computations can be reused to reduce computational costs. I cannot give a higher score because the work is still quite empirical due to limited task and model evaluations. While it sounds intuitive, the idea of leveraging reasoning redundancy is quite limited in practice because, as LLMs get stronger, they seldom produce repetitive words, leaving limited room for improvement. This can be seen in the authors' response to evaluating a different model, where on QwQ, the method achieved diminishing gains as the budget increased.

**Limitations:**

yes

**Quality:**

3

**Strengths And Weaknesses:**

Strengths:

1. **The core idea makes a lot of sense**. The paper points out that just looking at attention scores isn't enough in reasoning tasks—many tokens that look "important" are actually just repetitive. By explicitly addressing both importance and redundancy, the method targets a real gap in existing work.
2. **The method is clearly explained**. The paper does a good job of breaking down the algorithm into importance scoring, redundancy estimation, and joint selection. The decoding-time workflow—how tokens are kept or discarded—is easy to follow.
3. **The results are strong**. R-KV consistently outperforms SnapKV in both accuracy and efficiency. Getting close to or even beating full KV cache performance while saving up to 90% memory and boosting throughput is impressive, especially without any model retraining.

Weaknesses:

1. **The baseline comparison is a bit narrow**. The paper only compares against SnapKV, while other recent methods like StreamingLLM, H2O, QUEST etc. work also perform token pruning during decoding. Although these are discussed in the related work section, it would be more convincing to include at least some of them in the experiments. Given that they share similar motivations (e.g., reducing KV growth during decoding), experimental comparisons would help clarify where R-KV stands among these alternatives.
2. **All the experiments are on DeepSeek distilled models**. It’s unclear how well R-KV would generalize to other reasoning models like Qwen or QwQ. These models might generate differently, and redundancy patterns could vary.
3. **Efficiency analysis setup isn’t fully transparent**. It’s not clear what attention backend is used, and since the method doesn’t currently support things like paged attention in vLLM, it’s hard to know how useful R-KV is in production setups where those are standard. A discussion or test under those conditions would help.

---

> ### Author Rebuttal · Authors · 2025-07-31
>
> We thank the reviewer for the insightful feedback, and we would like to answer a few questions:
>
> > **Q1**: The baseline comparison is a bit narrow
>
> **A1**: As noted in Section 4.1, we chose SnapKV as the primary baseline because it directly uses attention scores for KV cache eviction. Since our method builds upon this paradigm by introducing a hybrid selection metric that combines attention with redundancy signals, SnapKV—representing the strongest pure attention-based KV cache eviction strategy—is the most appropriate point of comparison. As discussed in Section 4.1, our improvements specifically target and enhance this class of methods, making SnapKV a natural and focused baseline.
>
> We agree that including comparisons with other recent decoding-time KV pruning methods—such as StreamingLLM, and H2O—would provide a broader empirical perspective. To address this, we have conducted additional experiments comparing R-KV with several of these recent baselines. Results show that R-KV consistently outperforms them across budgets, highlighting its effectiveness in balancing memory reduction and performance preservation. We will include these results and their discussion in the revised version to strengthen the empirical scope of our work.
>
> **Deepseek-Distilled-Llama-8B - Math500**
>
> | Method                  | 128   | 256   | 512   | 1024  |
> |-------------------------|-------|-------|-------|--------|
> | StreamingLLM           | 21.8  | 29.4  | 42.8  | 54.4   |
> | H2O                     | 29.6  | 45.0  | 56.6  | 67.6   |
> |  SnapKV        | 32.53 | 50.07 | 64.03 | 74.43  |
> | R-KV          | 51.08 | 67.39 | 76.92 | 81.34  |
>
> > **Q2**: All the experiments are on DeepSeek distilled models
>
> **A2**: We tested R-KV on mathematical reasoning benchmarks using a different family of reasoning models—QwQ. R-KV consistently showed strong performance and effectiveness across these diverse reasoning tasks, highlighting its robustness and adaptability. Results are shown as below. We would include the results of the extended models in the revised version of our work.
>
> **QwQ**
>
> | - | MATH Pass@1 | - | AIME24 | - | AIME25 Pass@1 | - |
> |--------|------------------|--------------------|---------------------|-----------------------|---------------------|-----------------------|
> | budget | R-KV | SnapKV | R-KV | SnapKV | R-KV | SnapKV |
> | 128    | 0.4812           | 0.1522             | 0.0215              | 0.0021                | 0.0052              | 0.0011                |
> | 256    | 0.6476           | 0.334              | 0.0734              | 0.0063                | 0.0151              | 0.0031                |
> | 512    | 0.8031           | 0.5581             | 0.1479              | 0.0308                | 0.0802              | 0.0142                |
> | 768    | 0.8585           | 0.7476             | 0.2251              | 0.0604                | 0.162               | 0.0355                |
> | 1024   | 0.8897           | 0.8359             | 0.3062              | 0.1742                | 0.213               | 0.1088                |
> | 1536   | 0.9323           | 0.9032             | 0.4229              | 0.2833                | 0.2845              | 0.1891                |
> | 2048   | 0.9457           | 0.938              | 0.5354              | 0.4646                | 0.3396              | 0.2567                |
> | 2560   | 0.9559           | 0.9549             | 0.5729              | 0.5458                | 0.3718              | 0.3124                |
> | 3072   | 0.9623           | 0.9583             | 0.6014              | 0.5735                | 0.4033              | 0.3402                |
>
> > **Q3**: Efficiency analysis setup isn’t fully transparent
>
> **A3**: We used Flash-Infer and Flash-Attn as our attention backend during evaluation. In the revised version of our paper, we will also report the performance of R-KV when deployed in vLLM.

---

> > ### Comment · Reviewer_cogD · 2025-08-08
> >
> > Thanks for the clarification. I maintain my score.

---

### Official Review · Reviewer_evCh · 2025-07-03

**Clarity:** 3
**Significance:** 4
**Originality:** 4
**Rating:** 5
**Confidence:** 4

**Summary:**

This paper introduces Redundancy-aware KV Cache Compression for Reasoning models (R-KV), a method that computes and combines importance and redundancy scores for generated tokensthat guide their compression. The focus of R-KV is on decoding rather than prefilling stage with immediate application on the excessively long outputs of reasoning models. The core idea is managing two fixed size buffers for retained (budget buffer) and newly generated tokens and compressing between fixed-length text generation steps. In particular, a number of most recently generated tokens in the text are retained (observation tokens) and the rest tokens to fit in the budget buffer are selected by assigning scores to the remaining generated tokens and previously retained ones (candidate tokens). A token score is a linear combination of its importance (attention) score (where observation tokens serve as queries and a sliding window around a candidate token stabilizes the computation of the attention it receives) and its redundancy score (computed by averaging its key similarities to all other candidate tokens).

In the empirical evaluation of R-KV, authors typically report the accuracy on MATH500 and AIME 2024 datasets for various KV-cache budgets. Full KV-cache setup serves as the bare minimum baseline and SnapKV adapted to the same compression intervals R-KV is another (KV-cache compression) baseline; in most setups, models are Deepseek-R1 distilled ones (8B and 14B parameters). Reported results are impressive: in some configurations, R-KV can maintain the full KV-cache performance by using only 10% of the KV-cache, outperform it (still utilizing a fraction of the KV-cache) or achieve up to 90% memory savings and x4.9 speedup. Interestingly, the authors include a discussion on choosing the coefficients in the linear combination of the two scores and a nice illustration of the excessively hight scores other methods can put on very recent tokens or repetitive token subsets, however of minimal importance for eviction decisions.

**Questions:**

Can you elaborate a bit more on why choosing SnapKV as the only KV-cache compression method for comparison? There are many KV-cache compression methods, it is true that most focus on prefilling stage and need adaptation to be used in this context, but it would be nice to have some additional arguments. For example, where some other methods like H2O tried and they were found too unfit for this setup?

**Ethical Concerns:**

["NO or VERY MINOR ethics concerns only"]

**Final Justification:**

This work is timely, reported results are impressive. Also authors provided satisfactory clarifications and additional empirical evidence during rebuttal as requested. Assuming that presentation improvements will be integrated in the final version, I am maintaining my original positive score (5: accept).

**Limitations:**

Yes

**Quality:**

3

**Strengths And Weaknesses:**

+ Results are impressive and demonstrate that R-KV can be highly important in practice.

+ This work is timely, given that (output) reasoning traces are extremely lengthy and there is a pressing need for their KV-cache to compressed. Ideas from compressing the KV-cache of (input) long contexts can certainly be leveraged, but here the setup is more dynamic.

- There is room for improvement in the presentation: Basically some minor typos (e.g. in line 186 out of the three datasets mentioned only two are listed, in line 223-224 the reference should be to the 14B model) and most importantly the use of different symbols (alpha and lambda) for the coefficient in the combination of the two scores both in Figures and text (e.g. lines 240 and 243).

---

> ### Author Rebuttal · Authors · 2025-07-31
>
> We thank the reviewer for the insightful feedback, and we would like to address a few concerns:
>
> > **Q1**: There is room for improvement in the presentation.
>
> **A1**: We appreciate the feedback and will improve the presentation in the revised revision.
>
> > **Q2**: Can you elaborate a bit more on why choosing SnapKV as the only KV-cache compression method for comparison? There are many KV-cache compression methods, it is true that most focus on prefilling stage and need adaptation to be used in this context, but it would be nice to have some additional arguments. For example, where some other methods like H2O tried and they were found too unfit for this setup?
>
> **A2**: We chose SnapKV as the primary baseline because it directly uses attention scores for KV cache eviction. Since our method builds upon this paradigm by introducing a hybrid selection metric that combines attention with redundancy signals, SnapKV—representing the strongest attention-based KV cache eviction strategy—is the most appropriate point of comparison. As discussed in Section 4.1, our improvements specifically target and enhance this class of methods, making SnapKV a natural and focused baseline.
>
> We tried using StreamingLLM and H2O as baselines, but their performance was too poor and fell short of expectations. However, we agree that including comparisons with other recent decoding-time KV pruning methods—such as StreamingLLM, and H2O—would provide a broader empirical perspective. To address this, we have conducted additional experiments comparing R-KV with several of these recent baselines. Results show that R-KV consistently outperforms them across our budgets, highlighting its effectiveness in balancing memory reduction and performance preservation. We will include these results and their discussion in the revised version to strengthen the empirical scope of our work.
>
> **Deepseek-Distilled-Llama-8B - Math500**
>
> | Method                  | 128   | 256   | 512   | 1024  |
> |-------------------------|-------|-------|-------|--------|
> | StreamingLLM           | 21.8  | 29.4  | 42.8  | 54.4   |
> | H2O                     | 29.6  | 45.0  | 56.6  | 67.6   |
> | SnapKV        | 32.53 | 50.07 | 64.03 | 74.43  |
> | R-KV          | 51.08 | 67.39 | 76.92 | 81.34  |

---

> ### Comment · Reviewer_evCh · 2025-08-08
>
> I want to thank the authors for the additional empirical results and for clarifying the rationale for selecting SnapKV as the primary baseline. I am also satisfied by the expressed plans for improving the presentation. Therefore I maintain my positive score.

---

### Author Response · Authors · 2025-08-09
**General Response**

Dear Reviewers, ACs, SACs and PCs,

We sincerely appreciate your efforts in maintaining the review process and value the constructive, insightful feedback provided by all reviewers, along with your recognition of our work’s contributions. Below, we summarize the key strengths identified by each reviewer, clarify how we have addressed the raised concerns in our rebuttal, and outline planned improvements for the revised version.


## **Strengths Recognized by Reviewers**

### **Novelty and Motivation**
1. **Reviewer `evCh`** — Highlighted the practical importance and timeliness of R-KV for compressing lengthy KV caches in reasoning models, commending our effective use of attention + redundancy scoring to overcome pitfalls of pure attention-based eviction.
2. **Reviewer `cogD`** — Appreciated the clear problem statement and principled design of our joint scoring mechanism, emphasizing its relevance to a real gap in existing work.
3. **Reviewer `vf9G`** — Praised the strong motivation behind tackling redundancy in reasoning model outputs, calling R-KV a simple yet principled algorithm.
4. **Reviewer `Dks4`** — Noted the novelty of our redundancy-aware perspective, which directly addresses repetitive self-reflections overlooked by prior approaches.

### **Empirical Strength and Clarity**
1. **Reviewer `evCh`** — Found our results “impressive” and recognized our discussion on score coefficient selection and pitfalls in other methods.
2. **Reviewer `cogD`** — Pointed out that R-KV achieves near-parity or better than full KV cache accuracy while using only a fraction of the cache, delivering substantial throughput and memory gains.
3. **Reviewer `vf9G`** — Acknowledged the strong empirical results and practical deployability of our method.
4. **Reviewer `Dks4`** — Appreciated the comprehensive ablations, clear visualizations, and the lightweight, training-free nature of our approach.

## **Key Clarifications from the Rebuttal**

1. **Baseline Scope** — Addressing concerns from `evCh` and `cogD`, we expanded comparisons beyond SnapKV to include StreamingLLM and H2O, demonstrating R-KV’s consistent superiority across various budgets.
2. **Model and Task Generalization** — In response to `cogD`, `vf9G`, and `Dks4`, we added evaluations on the QwQ model and GPQA dataset (scientific QA), confirming robustness across models and domains.
3. **Hyperparameter Specification and Sensitivity** — As requested by `vf9G` and `Dks4`, we explicitly reported the β, and T values used in our experiments and added sensitivity analysis for β.

## **Planned Manuscript Revisions**

1. **Expanded Empirical Evaluation** — Incorporate additional baselines (StreamingLLM, H2O), domains (GPQA), and models (QwQ).
2. **Improved Presentation** — Correct typos and figure inconsistencies, and include further details on efficiency backends and hyperparameter documentation.
3. **Extended Analysis** — Provide quantitative redundancy analyses, discuss compatibility with other inference optimization techniques, and include sensitivity analyses for hyperparameters.


We believe these additions and clarifications will further strengthen the manuscript, making our contributions clearer, more generalizable, and more compelling. We are grateful for the reviewers’ detailed and thoughtful feedback, which has been invaluable in refining this work.

Best regards,
Authors

---

### Decision · Program_Chairs · 2025-09-17

**Decision:**

Accept (poster)

**Comment:**

**summary**
This paper introduces R-KV, a training-free method for compressing the Key-Value (KV) cache in large language models during inference on reasoning tasks. The core idea is to prune the cache by jointly considering both token importance (from attention scores) and token redundancy (from semantic similarity of key vectors). This dual-criterion approach aims to create a more information-dense cache, improving inference efficiency without sacrificing accuracy.

**strengths**
- The method's core idea of targeting redundancy in addition to importance is novel and addresses a clear limitation of prior attention-only eviction strategies.
- The empirical results are strong, demonstrating that R-KV can match or even exceed the performance of a full KV cache while using only a small fraction of the memory, leading to significant speedups.
- The approach is lightweight, training-free, and model-agnostic, making it a practical and easily deployable solution for accelerating LLM inference.

**weaknesses**
- The initial evaluation was limited in scope, focusing on a specific family of models and mathematical reasoning tasks, which raised questions about generalizability.
- The paper's initial baseline comparison was narrow, primarily focusing on SnapKV while omitting other relevant methods like StreamingLLM or H2O.
- The manuscript lacked a thorough analysis of key hyperparameters and their sensitivity, which is crucial for reproducibility and practical application.

**final descision**
The paper presents a novel, practical, and effective method for a significant problem. The strong empirical results, combined with a successful rebuttal that broadened the experimental scope, justify its acceptance.